

# vConTACT: an iVirus tool to classify double-stranded DNA viruses that infect *Archaea* and *Bacteria*

Benjamin Bolduc[1,*], Ho Bin Jang[1,*], Guilhem Doulcier[2,3], Zhi-Qiang You[4], Simon Roux[1] and Matthew B. Sullivan[1,5]

[1] Department of Microbiology, Ohio State University, Columbus, OH, United States
[2] Institut de Biologie de l'ENS (IBENS), École normale supérieure, PSL Research University, Paris, France
[3] ESPCI, PSL Research University, Paris, France
[4] Department of Chemistry and Biochemistry, Ohio State University, Columbus, OH, United States
[5] Department of Civil, Environmental and Geodetic Engineering, Ohio State University, Columbus, OH, United States

[*] These authors contributed equally to this work.

Corresponding author
Matthew B. Sullivan,
mbsulli@gmail.com,
mbsulli@email.arizona.edu

## ABSTRACT

Taxonomic classification of archaeal and bacterial viruses is challenging, yet also fundamental for developing a predictive understanding of microbial ecosystems. Recent identification of hundreds of thousands of new viral genomes and genome fragments, whose hosts remain unknown, requires a paradigm shift away from traditional classification approaches and towards the use of genomes for taxonomy. Here we revisited the use of genomes and their protein content as a means for developing a viral taxonomy for bacterial and archaeal viruses. A network-based analytic was evaluated and benchmarked against authority-accepted taxonomic assignments and found to be largely concordant. Exceptions were manually examined and found to represent areas of viral genome 'sequence space' that are under-sampled or prone to excessive genetic exchange. While both cases are poorly resolved by genome-based taxonomic approaches, the former will improve as viral sequence space is better sampled and the latter are uncommon. Finally, given the largely robust taxonomic capabilities of this approach, we sought to enable researchers to easily and systematically classify new viruses. Thus, we established a tool, vConTACT, as an app at iVirus, where it operates as a fast, highly scalable, user-friendly app within the free and powerful CyVerse cyberinfrastructure.

## INTRODUCTION

Classification of viruses that infect Archaea and Bacteria remains challenging in virology. Official viral taxonomy is handled by the International Committee for the Taxonomy of Viruses (ICTV) and organizes viruses into order, family, subfamily, genus and species. Historically, this organization derives from numerous viral features, such as morphology, genome composition, segmentation, replication strategies and amino- and nucleic-acid similarities—all of which is thought to roughly organize viruses according

to their evolutionary histories (*Simmonds, 2015*). As of 2015, the latest report issued, the ICTV has classified 7 orders, 111 families, 27 subfamilies, 609 genera and 3704 species (http://ictvonline.org/virusTaxInfo.asp).

Problematically, however, current ICTV classification procedures cannot keep pace with viral discovery and may need revision where viruses are not brought into culture. For example, of the 4,400 viral isolate genomes deposited into National Center for Biotechnology information (NCBI) viral RefSeq, only 43% had been ICTV-classified by 2015. This is because the lengthy 'proposal' processes lags deposition of new viral genomes, in some cases for years (*Fauquet & Fargette, 2005*). Concurrently, new computational approaches are providing access to viral genomes and large genome fragments at unprecedented rates. One approach mines microbial genomic datasets to provide virus sequences where the host is known—already adding 12,498 new prophages from publicly available bacterial and archaeal microbial genomes (*Roux et al., 2015a*) and 89 (69 and 20, respectively) new virus sequences from single cell amplified genome sequencing projects (*Roux et al., 2014*; *Labonté et al., 2015*). A second approach assembles viral genomes and large genome fragments from metagenomics datasets. The largest of such studies added 264,413 new putative (partial) viral genomes from fmicrobial and viral metagenomes across a broad range of ecosystems (*Paez-Espino et al., 2017*). Other studies include human stool samples (*Norman et al., 2015*; *Manrique et al., 2016*). Such new virus genomes and large genome fragments will keep coming for the foreseeable future and represent an incredible resource for viral ecology. While this opportunity is now clearly recognized in a recent Consensus Statement from the ICTV (*Simmonds et al., 2017*), it also represents a daunting challenge for taxonomy.

Currently such rapidly expanding genomic databases of the virosphere remain challenging to integrate into a systematic framework for three reasons. First, viruses lack a universal marker gene, which prevents the taxonomic starting place that is so valuable for microbes (*Woese, Kandler & Wheelis, 1990*). Second, though genomes and large genome fragments are now much more readily available, researchers are reticent to use genomes as a basis for taxonomy as a paradigm has emerged whereby viruses are rampantly mosaic and therefore must exist as part of a genomic continuum such that any clustering in 'sequence space' is an artifact of sampling. This is most well-studied in the many genomes of mycobacteriophages (*Pope et al., 2015*), but is contrasted by observations in cyanophages where efforts have been made to more deeply sample variability in a single site with findings suggesting clear population structure for naturally-occurring cyanophages (*Deng et al., 2014*) and that cyanophage populations appear to fit a population genetics-based species definition (*Marston & Amrich, 2009*; *Gregory et al., 2016*). It is possible that gene flow differs between DNA virus groups, depending upon their lifestyle. For example, lytic viruses spend very little time in a host cell (only long enough to lytically reproduce), whereas temperate viruses can spend generations replicating with its host cell as a prophage and during this time the prophage may be exposed to genomic sequence from super-infecting viruses and other mobile elements. The former lifestyle restricts these viruses to virus-host gene exchanges, except during co-infection, whereas the latter lifestyle would presumably enable more frequent virus-virus gene exchanges. As such, the lytic cyanophages might

maintain more discrete 'population' boundaries, while the more commonly temperate mycophages might exist as a continuum in sequence space due to higher rates of gene flow (*Gregory et al., 2016*; *Keen et al., 2017*). Thus, it remains unclear whether viral genomes can serve as the sole basis for taxonomy, or whether exploration of available data could help identify areas of viral genome sequence space that are amenable to taxonomic 'rules' and others that are not.

Despite these challenges, numerous reference-independent, automated, genome-based classification schemes for bacterial and archaeal viruses have been proposed. For these viruses, an early effort recognized that more genes are shared within related virus groups than between them (*Lawrence, Hatfull & Hendrix, 2002*), which led virologists to use translated genomes as the basis of whole genome phylogenomic tree classifications—e.g., the Phage Proteomic Tree (*Edwards & Rohwer, 2002*). Simulations showed this method to be very accurate for assigning fragmented reads to the correct genomes (*Edwards & Rohwer, 2005*) but it suffers from the availability of phage genomes. A second approach that has emerged for relatively well-studied virus groups, is to use pairwise distances between aligned sequences to identify discontinuities that can indicate classification thresholds. However, such approaches suffer from several issues: (i) they are not generalizable to the coming deluge of environmental viral genome sequences as they require *a priori* expert knowledge to impose similarity thresholds at each level, (ii) ICTV subcommittees have established varied sequence similarity thresholds across viral groups (*Simmonds, 2015*), which would require a sliding threshold, and (iii) the methods can only classify sequences that are similar to database references (*Zanotto et al., 1996*), which for the oceans at least represents <1% of the predicted viral genomes thought to exist (*Brum et al., 2015*).

Complementarily, two network-based approaches have been utilized to organize virus genome sequence space in a manner that enables classification without *a priori* knowledge. The first, a gene sharing network (*Lima-Mendez et al., 2008*), predicts viral genes in all the genomes, translates them into proteins, organizes these proteins into Markov cluster (MCL)-based protein families (protein clusters, "PCs"), evaluates the number of shared protein clusters pairwise throughout the dataset to establish a protein profile, and then represents this information as a weighted graph, with nodes representing viral genomes and edges the similarity score of their shared protein content. Given the 306 bacterial viruses (phages) known at the time, this method was precise as it correctly placed 92% and 95% of these phages into their correct ICTV genus or family, respectively (*Lima-Mendez et al., 2008*). A similar approach was used to assign a newly described phage to the phiKZ group (*Jang et al., 2013*). Since these genome networks use only one type of node, the graph is defined as monopartite (*Corel et al., 2016*). The second, a bipartite genome network consists of two distinct sets of nodes (i.e., protein families and genomes) with only links joining the nodes in different sets (*Corel et al., 2016*). Recently, all dsDNA viruses along with mobile genetic elements were analyzed with a bipartite approach, which revealed a module-based structure to the dsDNA virosphere (*Iranzo, Krupovic & Koonin, 2016*), while *Iranzo et al. (2016)* successfully extended the same network analytics to the archaeal viruses and related plasmids. Although both mono-/bipartite networks can be used as tools for investigating gene sharing across genomes, a bipartite graph directly displays the interactions between

**Table 1   Terminology used.**

| Terminology | Definition |
|---|---|
| Nodes | Also known as *vertices*, these are points within a network. In this work, they are viral genomes. |
| Edges | Also known as *arcs*, these lines connect nodes in the network. In this work, edges have a property called *weight*, which represents the strength (as measured by significance score) between two genomes. |
| Betweenness centrality (BC) | Measure of how influential a node is within a network, measured by the number of shortest paths that pass through the node from all other nodes. |
| Connected component | A subgraph in which any two nodes are connected to each other directly (to each other) or indirectly (through other nodes). |
| Largest connected component (LCC) | The connected component with the greatest number of nodes. |
| Viral cluster (VC) | A group of viral sequences sharing a sufficiently significant number of genes to not occur by chance between the genomes (as determined by the hypergeometric formula). |
| Protein cluster (PC) | A group of highly similar and related proteins, defined in this work using MCL on BLAST E-values between proteins. |
| Module Profile | A table-like representation of the presence/absence data between groups of protein clusters (modules) and groups of genomes (viral clusters). |
| Precision (P) | Also known as the *positive predictive value*, is a measure of how many true positives are identified. |
| Recall (R) | Also known as *sensitivity*, is a measure of how many of the total positives are identified. |

'gene families' and 'genomes', which are not depicted in a monopartite one (*Corel et al., 2016*). Thus, a bipartite approach can be more accurate in evaluating the gene sharing between and across genomes (*Iranzo, Krupovic & Koonin, 2016*; *Iranzo et al., 2016*). These two mono-/bipartite networks nonetheless imply that even very distantly related viruses can be organized into discrete populations by genomes alone and that there may be hope for automated, genome-based viral taxonomy, at least for dsDNA viruses.

Here we re-evaluated monopartite gene sharing networks and their efficacy for recapitulating ICTV-based classifications using an expanded dataset of 2,010 bacterial and archaeal virus genomes (available as of RefSeq v75), while also deeply exploring where network-based methods have lower resolution and/or yield discontinuities with currently established taxonomies. Further, we make these approaches accessible to researchers by developing a tool, vConTACT (Viral CONTigs Automatic Clustering and Taxonomy), and deploy it as part of the iVirus ecosystem of apps that leverages the CyVerse cyberinfrastructure (*Bolduc et al., 2016*).

# MATERIALS AND METHODS
## Terminology
Network topological parameters, their definitions and abbreviations are available in Table 1.

### Reference datasets

To test this methodology, we downloaded the entire NCBI viral reference dataset ("ViralRefSeq", version 75, containing 5539 viruses) and removed eukaryotic viruses by filtering against tables downloaded on NCBI's ViralRefSeq viral genome page (http://www.ncbi.nlm.nih.gov/genomes/GenomesGroup.cgi?taxid=10239). The resulting file ("Bacterial and Archaeal viruses"; BAV) contained 2,010 total viruses; 1,905 dsDNA, 88 ssDNA, 5 dsRNA and 12 ssRNA. All viruses contained taxonomic affiliation information, though not all viruses had affiliations associated with each level of the taxonomy (e.g., not all viruses have a "sub-family" designation). To improve taxonomic assignments, the ICTV taxonomy was also retrieved (https://talk.ictvonline.org/files/master-species-lists/) and the ICTV affiliations were used to supplement the NCBI data.

### Building protein cluster profiles

To generate sequence profiles with information about the presence or absence of a sequence within one or more protein clusters (described previously as protein *families* (*Lima-Mendez et al., 2008*), proteins from each sequence were first extracted from the ViralRefSeq proteins file. BLASTP (*Altschul et al., 1997*) was used to compare all proteins (198,102) from the sequences in an all-versus-all pairwise comparison (default parameters, except e-value 1E-5, bitscore 50). Protein clusters were subsequently identified using the Markov clustering algorithm (MCL) with an inflation value of 2, resulting in 23,022 protein clusters ("PCs"). Finally, we generated protein cluster profiles for each genome such that the presence of a gene within a protein cluster of a viral genome was given a value of "1" and the absence "0". This resulted in a large 2,010 × 23,022 matrix.

### Generating the similarity network

The similarity network is a graph where the nodes (i.e., reference sequences) are linked by edges when the similarity between their pc-profiles is considered sufficiently significant to not occur randomly. In other words, the network represents the overall similarity between sequences based on the number of shared protein clusters. To calculate the similarity between the profiles of two sequences (sequence $A$ and sequence $B$), the hypergeometric formula was used to estimate the probability that at least $c$ protein clusters would be in common:

$$P(X \geq c) = \sum_{i=c}^{\min(a,b)} \frac{C_a^i C_{n-a}^{b-i}}{C_n^b}. \tag{1}$$

Simply stated, the hypergeometric formula is used to calculate the probability that genomes A and B would have $c$ protein clusters in common by chance, which thus represents the statistical significance of an observed number of shared protein clusters between two genomes. The probability can be converted to an expectation value ($E$; for false positives) by multiplying the probability ($P$) by the total number of comparisons ($T$). The expectation value can then be converted into a significance score:

$$S(A,B) = -\log(E) = -log(P \times T). \tag{2}$$

Genome pairs with significance scores greater than 1 (i.e., E-value <0.1) are considered sufficiently similar (see permutation test, below) and were joined by an edge in the similarity network with a weight equal to their significance score. We refer to sequences within the network as *nodes*, the relationships connecting them, *edges* and the strength of that relationship, edge *weight*.

After generating the similarity network, groups of similar sequences (referred to as viral clusters, "VCs") were clustered by applying MCL with an inflation of 2.

## Measuring the proportion of shared genes between genomes

Given that genome sizes between pairs can differ greatly, this can lead to large differences in the proportion of the shared genes (*Ågren et al., 2012*). To counter this, we characterized the proportion of shared PCs between two genomes using the geometric index ($G$) as a symmetric index:

$$G_{AB} = \frac{|N(A) \cap N(B)|}{|N(A)| \times |N(B)|} \tag{3}$$

where $N(A)$ and $N(B)$ indicate the numbers of protein clusters (PCs) in the genomes of $A$ and $B$, respectively. This can provide a measure of the genome relatedness based on the percentage of conserved PCs between two genomes.

## Permutation test

The stringency of the significant score was evaluated through randomization of the original matrix where rows present viral genomes and columns PCs or singletons that are not shared with any other protein sequences (*Leplae et al., 2004*). Briefly, with an in-house R script, 1,000 matrices were generated by randomly rearranging PCs and/or singletons within pairs of genomes having a significant score $\leq 1$ (a negative control) and the scores associated with these random rearrangements were calculated. None of the genome pairs in this negative control produced significant scores >1, indicating values above this significance threshold did not occur by chance (*Lima-Mendez et al., 2008*).

## Affiliating sequence clusters with taxonomic groups

To assign (in the case of unknown sequences) or compare nodes (genomes) within clusters to their reference counterparts, we first defined *membership* of a node $c$ to a cluster $k$ $B(c,k)$ according to two methods, conservative and permissive. The conservative method (4) directly takes the result from the MCL clustering to assign a node to a cluster:

$$B(c,k) = \begin{cases} 1 \text{ if Contig } c \in \text{Cluster } k, \\ 0 \text{ otherwise} \end{cases} \tag{4}$$

while the permissive method takes the sum of all edge weights $w$ linking the node to nodes of the cluster, with the node becoming a member of its maximal membership cluster (5):

$$B'(c,k) = \frac{\sum_{i \in k} w_{c,i}}{\sum_{p \in \{\text{Clusters}\}} \sum_{j \in p} w_{g,j}}. \tag{5}$$

The precision $P(k,t)$ of the taxonomic class $t$ with respect to a cluster $k$ was defined as the proportion (in membership) of reference contigs of class $t$ in the membership of reference

contigs in the cluster $k$.

$$P(k,t) = \frac{\sum_{\forall i \in \{\text{sequence of class } t\}} B(i,k)}{\sum_{\forall j \in \{\text{reference sequence}\}} B(j,k)}. \qquad (6)$$

A cluster and all its node members are then affiliated with its maximal precision class. For the conservative method, the cluster is affiliated with the taxonomic class associated with the majority of its members. In cases where clusters do not contain at least half reference sequences, the entire cluster will be unaffiliated.

## Measuring the connectivity of genomes to clusters

The connection strength of a node $g$ to cluster $c$ was calculated as the average edge weight linking it to nodes of cluster $c$:

$$W_{g,c} = \frac{1}{k} \sum_{i=1}^{k} wg,i \qquad (7)$$

where $k$ and $w$ are the number and total weight of edges of the node $g$ in the cluster $c$, respectively. We refer to the average edge weight for node $g$ to the cluster it belongs to as its in-VC average weight, and to other clusters within the network as out-VC average weight.

## Identifying sub-clusters

To further subdivide heterogeneous clusters (those comprising $\geq 2$ taxa), cluster-wise module profiles (i.e., a module profile only including viruses previously identified as belonging to the same viral cluster) were hierarchically clustered using UPGMA with pairwise Euclidean distances implemented in Scipy.

## Statistical calculations

All calculations, statistics, network statistical analyses were performed using in-house python scripts, with the Numpy, Scipy, Biopython and Pandas python-packages. vConTACT is implemented in python with the same dependencies. The tool is available at https://bitbucket.org/MAVERICLab/vcontact. Scripts used in the generational and calculations of data are available at https://bitbucket.org/MAVERICLab/vcontact-SI.

## Network visualization and analysis

The network was visualized with Cytoscape (version 3.1.1; http://cytoscape.org/), using an edge-weighted spring embedded model, which places the genomes or fragments sharing more PCs closer to each other. Topological properties were estimated using a combination of python and the Network Analyzer 2.7 Cytoscape plug-in (*Assenov et al., 2008*).

# RESULTS AND DISCUSSION

## vConTACT analytical workflow and terminology

The vConTACT analyses are based on previously established gene sharing network methods (*Lima-Mendez et al., 2008*). Briefly, PCs are established across all genomes in the dataset; with vConTACT doing this by default using MCL clustering from all-versus-all BLASTP comparisons (though user-specified clusters can also be used). PC *profiles* of genomes

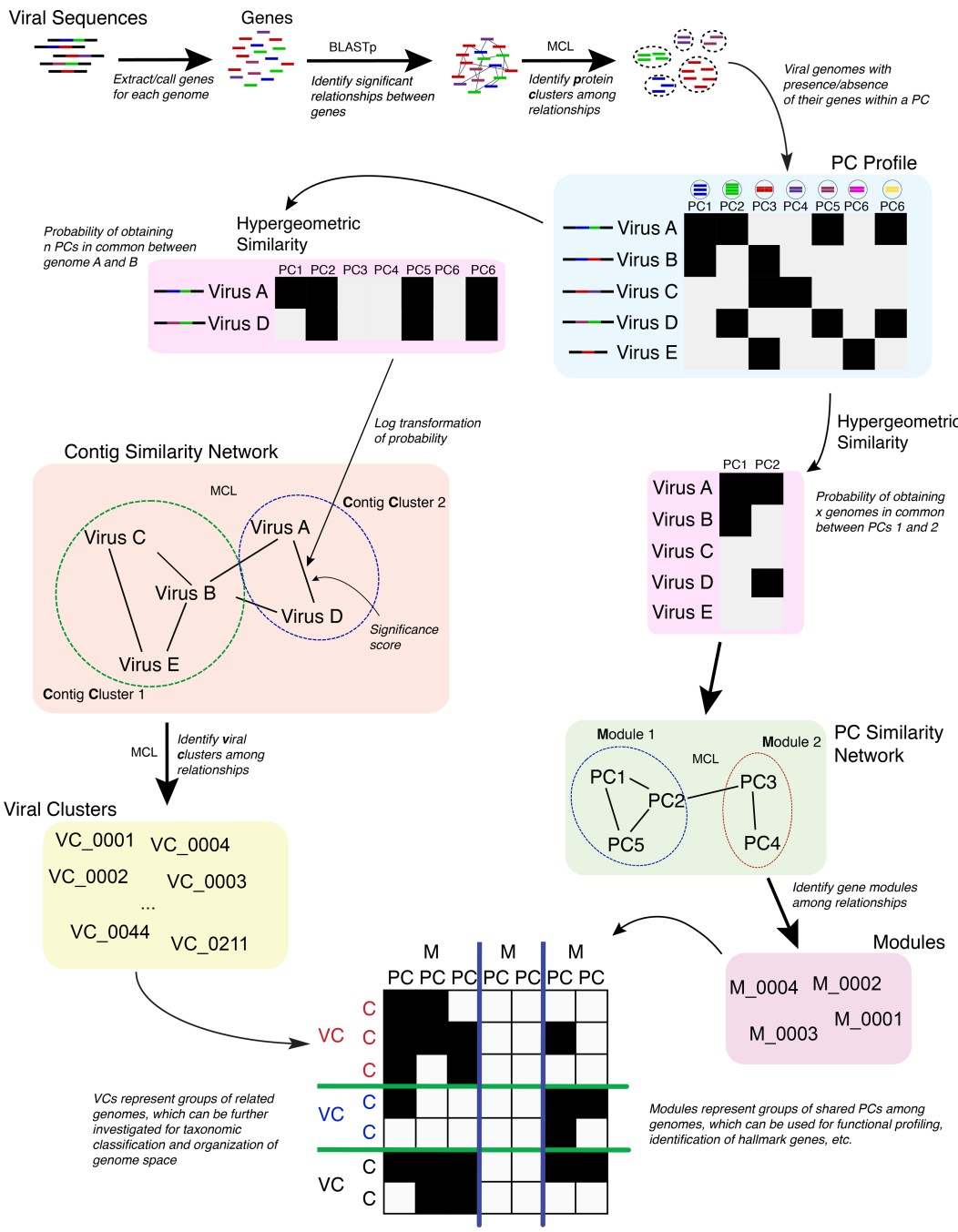

**Figure 1** Overview of the vContact processing pipeline.

or genome fragments (herein 'genome') are then calculated, where the presence and absence of PCs (from the entire PC dataset) along a genome are established and then compared pairwise between genomes (Fig. 1). The pairwise genome comparisons are then mathematically adjusted (using the hypergeometric similarity formula) to establish a probability that any genome pair would share $n$ PCs, given the total number of all PCs. This probability is log-transformed (in similar fashion to BLAST E-values) into a

significance score and applied as a *weight* to an edge between the two paired genomes in a similarity network. High significance scores represent a low probability that two genomes would share *n* PCs by chance, which can be interpreted as evidence of gene-sharing and presumably evolutionary relatedness between the paired genomes. After evaluating all pairings in the dataset, significance scores ≥1 are retained, and a network of the remaining genome pairs is constructed. MCL is subsequently applied to identify structure in the gene sharing network, but now the clusters represent groups or related genomes and are termed viral clusters ("VCs"). MCL is also applied against the network of PCs, whose members can be similar to members of other PCs. This effectively organizes the PCs into a higher-order structure known as a *protein module*. The relationship information identified from the genomes (organized into VCs) and PCs (organized into protein modules) are used to create a *module profile*, which can then be mined for taxonomic identification, functional profiling, etc.

## Benchmarking network-based taxonomy

To benchmark the ability of network-based taxonomy to capture 'known' viral relationships, we evaluated how vConTACT "re-classified" viral sequences at various taxonomic levels using 2,010 bacterial and archaeal viral genomes from VirRefSeq (v75). Of these reference genomes, ICTV-classifications were only available for a subset; 654 viruses from 2 orders, 738 viruses from 19 families, 152 viruses from 11 subfamilies, and 562 viruses from 158 genera. The network was then decomposed into VCs (described above) and a permutation test was used to establish significance score thresholds to prevent random relationships from entering the network. This analysis used the initial network's edge information to construct a matrix between genome pairs, and then permuted the edges 1,000 times. No edges were found to be significant during these tests, suggesting that relationships seen within the network did not arise by chance and could be confidently used to establish taxonomic groupings (see 'Materials and Methods', Table S1).

The resulting network, consisting of 1,964 viruses (nodes) and 65,393 relationships (edges, Fig. 2A), was then used as a basis for comparison to the ICTV-based classifications. Forty-six singleton viruses that do not have close relatives (2.2% of the total virus population) were excluded. A total of 211 VCs were identified, spread among 46 components (unconnected subnetworks), which more than doubles the 17 connected components identified previously (*Lima-Mendez et al., 2008*). Of the 46 components, 38 included 1,891 phages representing 194 VCs (left, Fig. 2A), and 8 components included 73 archaeal viruses representing 17 VCs (right, Fig. 2A). Most (87%) of the 1,891 phages belonged to the order *Caudovirales*, and comprised the largest connected component (LCC) in the analysis (top left, Fig. 2A). At the VC level, the network clustering performed well with average (across each taxonomic level) recall/precision percentages of 100%/100%, 90%/86%, and 80%/80% at the order, family and genus levels, respectively (Fig. 2B). Of the 211 VCs resolved by the network, 76.4% contained a single ICTV-accepted genus, suggesting a large concordance between the network VCs and accepted taxonomy, whereas

Peer J

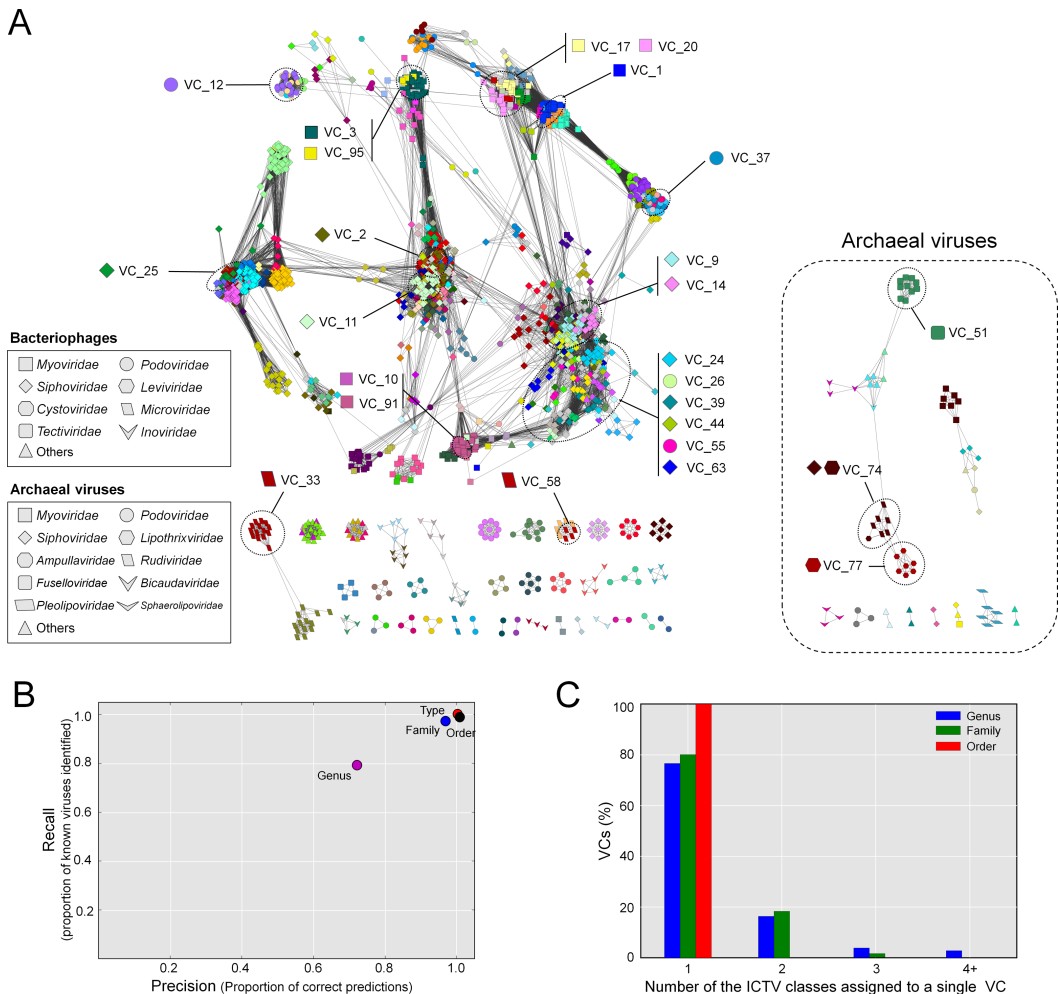

**Figure 2** **Protein-sharing network for 1,964 archaeal and bacterial virus genomes benchmarked against ICTV-accepted viral taxonomy.** (A) Each node represents a viral genome from RefSeq, with its shape representing the viral family (as indicated in the legend) and each distinct color the node's viral cluster (VC). Edges between nodes indicate a statistically significant relationship between the protein profiles of their viral genomes, with edge colors (darker = more significant) corresponding to their weighted similarity scores (threshold of $\geq 1$). VCs within the network are discriminated using the MCL algorithm ('Materials and Methods') and denoted as separate colors. The position of 26 heterogeneous VCs that contain 2 or more genera is indicated. (B) Precision and recall of network-based assignments as compared to ICTV assignments for each taxonomic level (genus, family, order, and type). (C) Percentage ($Y$-axis) of VCs that contain the number ($X$-axis) of each ICTV taxonomic level (genus, family, and order).

15.1% and 8.5% of the VCs contained two and 3 or more genera, respectively (Figs. 2A and 2C). Thus, roughly 3 out of 4 of the VCs cleanly correspond to ICTV genera.

Mechanistically, these discrepancies between network clustering and the ICTV classification could derive from either (i) under-sampling such that VCs with fewer members may not represent the naturally-occurring diversity of that viral group, or (ii) genetic exchanges between viral genomes that blur taxonomic boundaries between VCs.

To discriminate between these possibilities, we investigated further these "ICTV-discordant" areas of the network containing 2 or more ICTV genera (referred to as *heterogeneous VCs*), focusing on three of the more well-populated (many member genomes) heterogeneous VCs, and the archaeal virus heterogeneous VCs, which are among the least well-sampled taxa. Of the well-sampled VCs, VCs containing the 2nd, 3rd, and 4th most members (i.e., genomes), included the following: (i) VC1 contains the 8 genera belonging to the *Tevenvirinae* subfamily (*T4virus*, *Cc31virus*, *Js98virus*, *Rb49virus*, *Rb69virus*, *S16virus*, *Sp18virus*, and *Schizot4virus*) and a genus of the *Eucamyvirinae* (*Cp8virus*), as well as the *Tg1virus* and *Secunda5virus* that are not assigned to a particular subfamily, (ii) VC2 contains three genera (*Biseptimavirus*, *Phietavirus*, and *Triavirus*) belonging to the *Siphoviridae* family, and (iii) VC3 contains four genera (*Kayvirus*, *Silviavirus*, *Twortvirus*, and *P100virus*) belonging to the *Spounavirinae* of the *Myoviridae* and the six *Bacillus* virus genera (*Agatevirus*, *B4virus*, *Bc431virus*, *Bastillevirus*, *Nit1virus*, and *Wphvirus*) belonging to the *Myoviridae*. Finally, among the 73 archaeal viruses, only the *Fuselloviridae* were accurately classified at the genus level, while most (63%) archaeal viruses were incorrectly classified at the genus level.

## Gene content analyses suggest ICTV classifications should be revised for well-sampled taxa

A total of 23.6% of the VCs contained genomes from ≥2 ICTV-recognized genera, which suggests 'lumping' by the network analyses (via MCL) or 'splitting' during ICTV classification. To assess this, we computed the fraction of PCs that were shared both within an ICTV genus and between the multiple ICTV genera found in each heterogeneous VC and represented them as the percentage of intragenus similarity and intergenera similarity, respectively. Of the 25 VCs, intragenus similarities of all but one (VC9) shared more than 40% of their PCs (Fig. 3A, Table S2), which is consistent with the threshold commonly used to define a new dsDNA viral genus (*Lavigne et al., 2009*). In contrast, the intergenera similarities varied widely—some VCs (VCs 1–3, 9–11, 17, 20, 25, 33, 58, 91, 95) shared 20–40% of their PCs (subfamily level), whereas others shared more than ~40% (VCs 12, 14, 24, 26, 37, 44, and 51) or less than ~20% (VCs 39, 55, 63, 74, and 77) of their PCs. Where intergenera similarities are high (>40% of the PCs are shared), there may be a case to be made for merging the currently recognized ICTV genera. Consistent with this, all 6 of these highly (>40%) similar VCs (12, 14, 24, 26, 37 and 51) are suggested to be in need of revision, as these include *G7cvirus*, *N4virus*, *T1virus*, *Hp34virus*, and *Phikmvvirus* (*Wittmann et al., 2015*; *Eriksson et al., 2015*; *Niu et al., 2014*; *Krupovic et al., 2016*). Additionally, we found that in VC44, the phage CAjan, belonging to the *Seuratvirus*, shared 41.6–42.7% of its genes with three phages (JenP1 and 2 and JenK1 of the *Nongavirus* (Table S2)). Where intergenera similarities are lower (<20%, or 20–40% of the PCs are shared), the appropriate taxonomic assignment may require deeper sampling of viral genome sequence space and/or further network analytic development.

To further assess these cases, we next examined four VCs (1–3, 14) that contained more than 4 ICTV-recognized genera using hierarchical clustering of PC presence-absence data for each genome (Fig. 3B). In parallel, we computed the actual connectivity of the

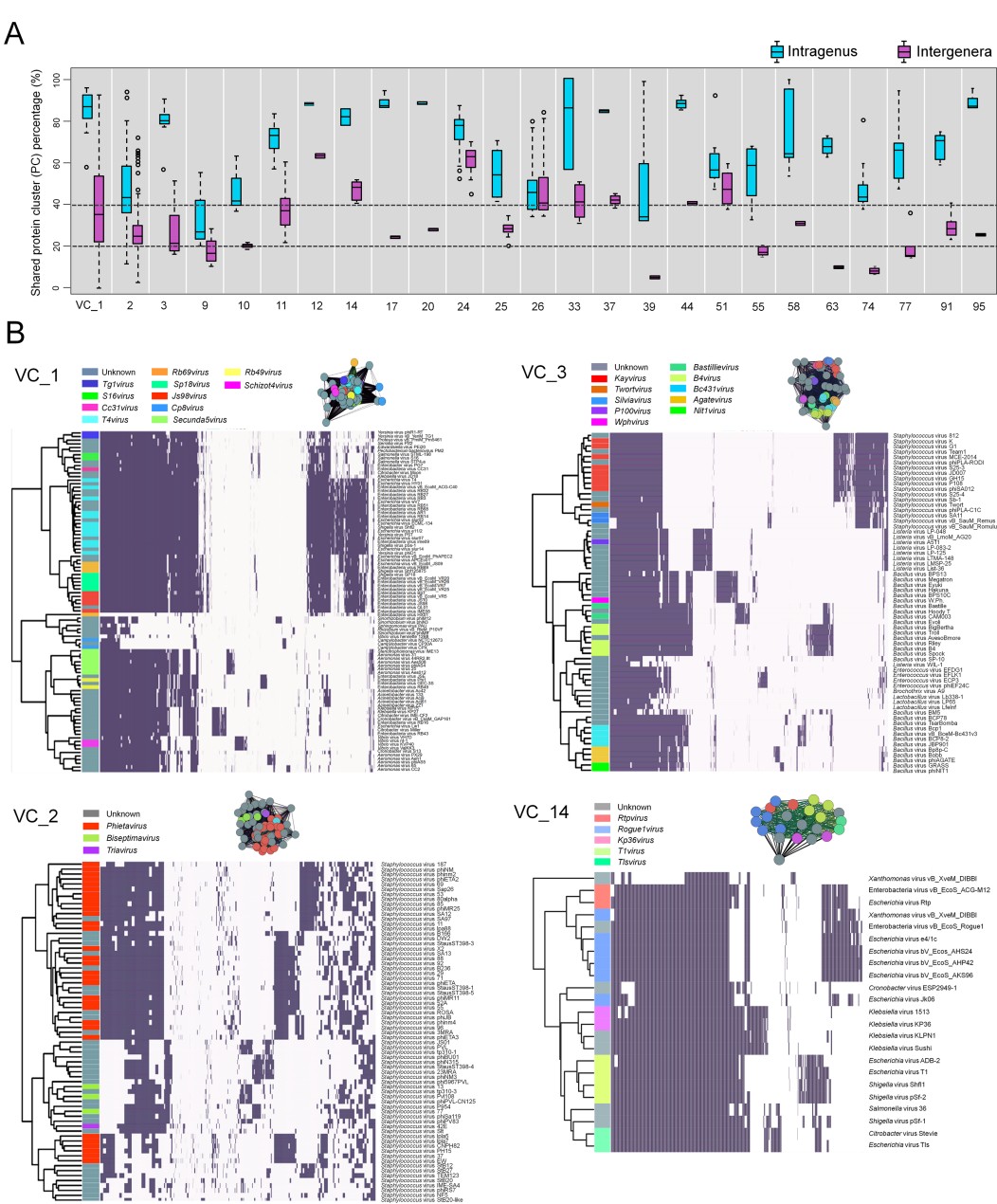

**Figure 3  Heterogeneous VCs.** Evaluation of VCs which contained taxon representatives from more than one ICTV genus. (A) Box plots show the percent inter- and intra-genus proteome similarities in the heterogeneous VCs. Dotted lines indicate the cut-off values of 20% and 40% proteome similarities to define the subfamily and genus, respectively, which have been ratified by the ICTV Bacterial and Archaeal Viruses Subcommittee. (B) Module profiles showing the presence and absence of PCs across genomes. Presence (dark box) denotes a gene that is present within a protein cluster. Genes from related genomes often cluster into the same PC, with alignments of highly related genomes showing large groups of PCs. Genomes are further partitioned using hierarchical clustering (see 'Materials and Methods').

genomes within these heterogeneous VCs according to the average weight of edges that (i) are between genomes of the same VC (in-VC avg. weight) and (ii) between the genomes of other VCs (out-VC avg. weight) (Table S3; 'Materials and Methods'). For example, within VC1, 8 genera of the *Tevenvirinae* (*S16virus*, *Cc31virus*, *T4virus*, *Rb69virus*, *Sp18virus*, *Js98virus*, *Rb49virus* and *Schizot4virus*) and their relatives (*Tg1virus* and *Secunda5virus*) share, on average, 61% and 38% of their total PCs, respectively, and 39% between all 10 genera (Table S2). Outside VC1, they share ∼11.2% of genes with other viral groups (Table S2). We found that the 10 genera within VC1 are more tightly interconnected than those of the 210 VCs overall, with average in-cluster values of 223.7 and 131.9 and average out-cluster values of 13.1 and 9.0, respectively (Table S3). These observations indicate that higher cross-similarities of 10 genera can be attributed to a large fraction of their shared genes, whereas only a small fraction of gene shared by other groups can hold them together.

Upon closer inspection, some of this 'lumping' appeared to be due to poorly sampled regions of sequence space. For example, VC1 also contained the *Cp8virus* of the subfamily *Eucampyvirinae*, which is odd to be placed alongside the *Tevenvirinae*, given that the other ICTV-recognized genus (*Cp220virus*) of the *Eucampyvrinae* is grouped into a separate cluster (VC 87). Since both genera (*Cp8virus* and *Cp220virus*) are distantly related to the *Tevenvirinae* (Javed et al., 2014), displaying only ∼11% shared genes to other *Tevenvirinae* (an average weight of 18.5) and ∼6% (11.8), respectively (Tables S2 and S3), these groupings might be driven by the fact that only 2 reference genomes (i.e., *Campylobacter* phages CPX and NCTC12673) are available in our ViralRefSeq dataset for *Cp220virus*. To test this, we artificially doubled the number of the genomes for this group by adding their replicas (phages CPX_copy1 and NCTC12673_copy1, Table S4) to the network. For all edges between the replicas and original genomes and outside them, vConTACT recalculated the weights. This led the *Cp220virus* genomes to clearly separate from VC1 and instead be correctly placed alongside VC 87 (Table S4). Consistently, among the heterogeneous VCs 39, 55, 63, 74, and 77 showing <∼20% intergenera similarities (Figs. 3A and S1), increasing the genome numbers of poorly-sampled ICTV genera led to clustering of members of those genera into their correct VCs (Table S4). Together these findings suggest that additional sampling in poorly sampled areas of viral sequence space will be required to most accurately establish genome-based taxonomy—issues that parallel those presented by long branch attraction for phylogenies (*Bergsten, 2005*).

Similar structure emerged from hierarchical clustering of PC presence/absence data from the 3 other well-represented heterogeneous VCs. In VC2, the three known subgroups of the *Phietavirus* (*Gutiérrez et al., 2014*) were resolved, sharing 44.9% of their PCs, and separate from two other subgroups—the *Biseptimavirus* and *Triavirus*, which shared 22.3% of their PCs (Fig. 3B, Table S2). A detailed analysis of VC2 revealed that phages phinm4 and 88, and phiETA2, 53, and 80alpha, belonging to subgroups 1 and 2 of the *Phietavirus*, respectively, and phage 77 from the *Biseptimavirus* share 35.6% to 43.8% of total PCs (Table S2), which straddles the genus boundary (*Lavigne et al., 2009*). Along with these six phages, other members of the *Phietavirus* and *Biseptimavirus* share ∼25% of their PCs (Table S2). The considerable fraction of shared PCs between the *Phietavirus* and *Biseptimavirus* argues for their lumping into the same cluster. Notably, despite the evolutionary relationship of

*Staphylococcus* phage 42e to the *Triavirus* (*Gutiérrez et al., 2014*), we found it is included into VC2, and separated from VC38 that exclusively consists of four members (phages 3A, 47, Ipla35, and Phi12) of the *Triavirus* (Table S3). Comparison of their connectivities reveals that, relative to the four *Triavirus* members within VC38 (avg. weight of 118.27; avg. shared PCs of 72.3%), phage 42e show weaker connections to VC38 (77.63; 49.5%) (Tables S2 and S3). This relationship is somewhat similar to the whole-genome phylogenetic tree of the *Triavirus* where four members of the *Triavirus* are more closely related to each other than to phage 42e (*Gutiérrez et al., 2014*). Further, phage 42e shows stronger connections to VC2 (33.59; 25.7%) than those of four *Triavirus* members (18.94; 17.9%) (Tables S2 and S3). Thus, given the drawback of MCL that cannot efficiently handle modules with overlaps (*Nepusz, Yu & Paccanaro, 2012*; *Shih & Parthasarathy, 2012*), phage 42e appears to be spuriously assigned to VC2 due to its highly-overlapped genes between VCs 2 and 38.

In VC3, containing the *Spounavirinae* (*Krupovic et al., 2016*), each sub-cluster has a corresponding ICTV genus with largely overlapping sets of genes while also showing a clearly distinct set(s) of genes. Of these, the six *Bacillus* virus genera (*Wphvirus*, *Bastillevirus*, *B4virus*, *Bc431virus*, *Agatevirus*, and *Nit1virus*) appear to be closely related to the *Spounavirinae*, with ~20% of total PCs in common (Fig. 3B, Table S2). Additional comparisons of the connectivities of clusters revealed that 10 genera of VC3 form strong connections to each other, but weak connections with the rest of network (in-and out-VC avg. weights of 118.16 and 14.54, respectively; Table S3). Thus, despite the fraction of genes specific to each genus (Fig. 3B), these high interconnectivities of 10 genera can join them together, which is similar to VC1. Finally, VC14 produced a clear division of the *Tunavirinae* (*Krupovic et al., 2016*), in which the *Escherichia* virus Jk06 is placed in a separate branch due to its less shared common genes (~56%) to the other *Rogue1virus* members (~82%); their highly-overlapped genes between genera above the genus boundary (40%) are associated with "taxonomic lumping" as described above (*Niu et al., 2014*; *Krupovic et al., 2016*).

We next evaluated three phage groups which were poorly represented in the S277 network (*Lima-Mendez et al., 2008*) and also represent some of the most abundant, widespread, and/or extensively studied phage groups (*Grose & Casjens, 2014*; *Pope et al., 2015*; *Roux et al., 2015b*)—the mycobacteriophages, *Tevenvirinae*, *Autographivirinae* and the archaeal viruses.

## Mycobacterium phages

The largest viral group covering 16.1% of the total population of the LCC (mostly *Caudovirales*, top left Fig. 1A) includes phages infecting *Mycobacteria*. The 318 mycophage genomes were assigned to 14 VCs (Fig. 4A), 13 of which were composed of reference genomes belonging to a single ICTV-recognized genus for each VC. The 14th mycophage VC, VC25, contained three ICTV-recognized genera—the *Bignuzvirus*, *Charlievirus*, and *Che9cvirus*. Although the module-based approach discerned the structure in this VC, which would group them into the known genera (Fig. S1), this "lumping" into a single VC reflects (i) their undersampling (i.e., each genus has 1 to at most 3 viruses) and/or (ii) highly-overlapped genes between genera. Indeed, of the 3 phages belonging to the

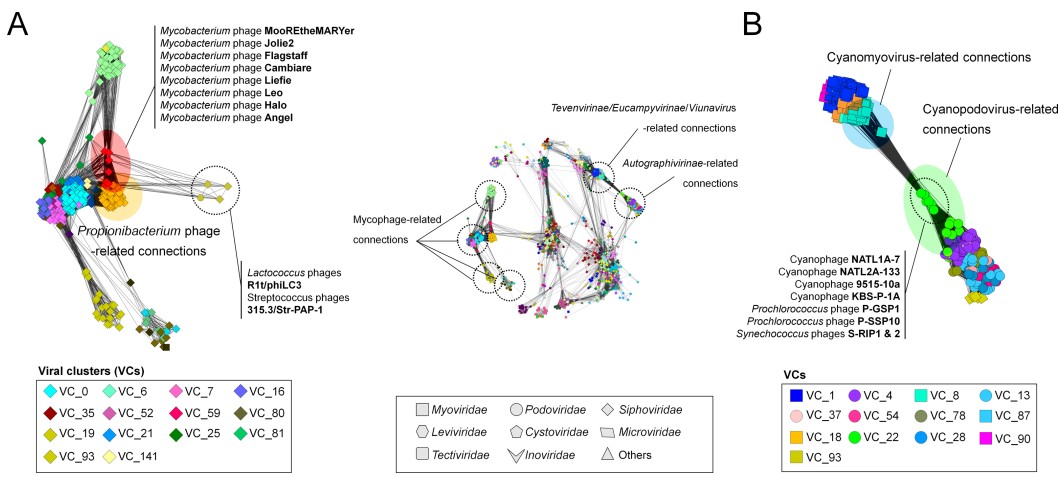

**Figure 4** **A detailed view of network regions containing three major viral groups and their relatives.** Viruses (nodes) are grouped by the MCL clustering. Each node in (A) and (B) is colored according to the viral cluster (VC) to which the corresponding virus belongs, which is shown in the legendary box in (A) and (B) respectively. Nodes are depicted as different shapes, presenting viruses belonging to the family of a given ICTV class or uncharacterized and others (legendary box between A and B). The location of viral groups is indicated for illustrative purposes.

*Che9cvirus*, phages Babsiella and Che9c shared 45% of their genes, but also shared 35% and 36% of their genes with the *Bignuzvirus* and 28% and 32% with the *Charlievirus*, respectively (Table S2), which results in higher connectivity between three genera than to other viral groups (Table S3). These findings contrast those in the rest of the network, and suggest that some phage groups (e.g., mycophages) may more frequently exchange genes than others.

To quantify this, we next examined features of the network reflecting the rate of gene sharing across viruses. Among 14 mycophage-related VCs, 12VCs (∼86%) appeared to form a densely connected region with variable edge weights (Fig. 4A; Table S3). For example, nine VCs including VCs 0 (*L5virus*), 7 (*Che8virus*), 16 (*Cjw1virus*), 21 (*Tm4virus*), 25 (*Bignuzvirus*, *Charlievirus*, and *Che9cvirus*), 52 (*Omegavirus*), 59 (*Liefievirus*), 112 (*Corndogvirus*), and 141 (taxonomically-unknown) were highly interconnected to each other, with weights of 1.1 to 21.2 (Table S3). Of these, VCs 16, 21, and 52 additionally linked to VC35 (*Bronvirus*). VC80 (*Barnyardvirus*) linked to VC81 (*Pbi1virus*). These web-like connections of mycophage-related VCs (or genus) strongly suggests that their genomes may be prone to frequent gene exchanges across taxonomic boundaries, supporting the previous finding of genomic continuity of mycophage populations (*Pope et al., 2015*), and consistent with the largely temperate phage lifestyle of the mycophages.

Of these mycophage VCs, many VC59 mycophages were broadly linked to nine VCs that contain other mycophages and phages from diverse hosts (Fig. 4A). To characterize this further, we analyzed the topological properties using the betweenness centrality (BC), which can identify the node residing in the shortest path between two other nodes (*Halary et al., 2009*). Specifically, in the shared-gene network, high-betweenness nodes (phages) can act as bridges between phages that would remain disconnected, due to their mosaic

content of genes (*Lima-Mendez et al., 2008*). Indeed, these eight VC 59 phages had 42-fold higher average BC than those of other mycophages and their relatives (0.04 vs. 9.45E−04) (Fig. S2).

However, this BC-based detection of mosaic viruses in monopartite network could be limited by the lack of identification of the genes responsible for these genomes connections. For example, based on the betweenness value, *Lima-Mendez et al. (2008)* identified a single representative of T5-like phages (i.e., a phage T5) as a mosaic virus bridging T4-/lambda-like phages. Recently, however, *Iranzo, Krupovic & Koonin (2016)* specified viral core genes and subsequently found that the bridge location of a phage T5 between T4-/lambda-like phages could arise from (i) the incomplete sampling of the *T5virus* and/or (ii) widespread viral hallmark genes having no obvious ancestors. Thus, in a monopartite network, BC values would have to be considered alongside the list of PCs associated with each edge to correctly identify mosaic viruses.

## The *Tevenvirinae*

As the second-largest group, containing 94 viruses in the heterogeneous VC1, which were further connected to 74 distant relatives and taxonomically unclassified myo-/siphovirus(s), the *Tevenvirinae* appeared to be restricted to a densely interconnected region (Fig. 4). A subsequent hierarchical clustering within VC1 grouped these 168 viral genomes into 5 subgroups (Fig. S3). Interestingly, three phages infecting cyanobacteria (P-SSM2, P-SSM4, and S-PM2) and T4-like phages that were initially found in a single cluster (*Lima-Mendez et al., 2008*) are separated into two clusters: VC8 containing the Exo T-evens and VC1 containing the T-evens/Pseudo/Schizo T-evens, respectively (*Filee, 2006*) (upper in Fig. 4B; Fig. S3). This network grouping can thus correctly identify the specificity of the Exo T-evens, including cyano- and pelagiphages, which the literature suggests to be only distantly related to other T4 superfamily viruses (*Comeau & Krisch, 2008*; *Roux et al., 2015b*).

## The *Autographivirinae*

We further identified 8 VCs associated with the *Autographivirinae*. Of four genera defined by the NCBI and/or ICTV, the *T7virus*, *SP6virus*, *Kp34virus* were found in VCs 4, 28, and 37, respectively, whereas the *Phikmvvirus* were spread across VCs 13 and 37 (Fig. 4B; also Fig. S4). Notably, a previous phylogenetic study based on three conserved proteins (i.e., RNA polymerase, head-tail connector and the DNA maturase B) showed considerable diversity of the *phikmvvirus* (*Eriksson et al., 2015*). We also observed distinct patterns of PC sharing between the PhiKMV-related genome(s) and other viruses in each cluster (Fig. S4), suggesting that the *Phikmvvirus* should likely be divided into two new subgroups.

In addition, among the recently emerged groups, nine *Acinetobacter* phages (*Huang et al., 2013*), as well as phage vB_CsaP_GAP227 (*Abbasifar et al., 2013*) and its close relatives were found in VCs 54 and 93, respectively (Fig. S4); all of them encode T7-specific RNA polymerase (*Lavigne et al., 2009*), which suggest that they fall within the *Autographivirnae* subfamily.

## Cyanophages

Many viruses are now thought to co-opt host genes to improve viral fitness; these stolen 'auxiliary metabolic genes' (AMGs) are well known from cyanophage genomes (photosynthesis genes; *Sullivan et al., 2006*; *Millard et al., 2009*; *Labrie et al., 2013*), but also from ocean viral metagenomes where viruses are now shown to contain genes involved in central carbon metabolism (*Hurwitz, Hallam & Sullivan, 2013*) and nitrogen and sulfur cycling (*Roux et al., 2016*) in ways that likely drive niche differentiation (*Hurwitz, Brum & Sullivan, 2014*). Thus, it is striking that VC22 in our network, which contains 19 cyanopodoviruses, had many linkages to taxonomically disparate *Tevenvirinae*, which turned out to be driven by photosynthesis genes shared across these viral taxa (Fig. 4B). Such "host" genes in viruses can bring taxonomically disparate viral groups closer together, and the network can thus help identify such niche defining viral genes for viruses infecting well studied hosts.

A recent phylogenomic analysis of 142 cyanomyoviruses found that these viruses can be split into multiple lineages, but most of the viral lineages have evolved to maintain their structures (*Gregory et al., 2016*). They additionally suggested that the contrasting pattern of gene flow between cyanophages and mycophages could be due to their lifestyle, i.e., lytic cyanomyoviruses and temperate mycophages, but this conclusion is based on a currently-limited collection of sequenced viral genomes. We also observed that a total of 74 cyanophages exclusively belong to VCs 8 (cyanomyoviruses) and 22 (cyanopodoviruses) with limited connections outside them (Fig. 4B; Table S2), which is different from reticulate inter-cluster (or genus) relationships of mycophage populations (discussed above), and suggests that among cyanophages the predominately lytic lifestyles restrict gene flow between viruses to presumably less common co-infection events.

## The archaeal viruses

Of the 72 archaeal viruses, 66 were associated with 18 VCs, while 6 viruses (Haloviruses HHTV-1 and VNH-1, Hyperthermophilic Archaeal Virus 1 & 2, Pyrococcous abyssi virus 1, and His 1 virus) were not included in the network, due to lack of statistically significant similarity to any other virus. Of the 25 heterogeneous VCs, archaeal viruses comprise 3 of them (VCs 51, 74 and 77), likely owing to their gene products showing little similarity to published viruses outside of other archaeal viruses (*Prangishvili, Garrett & Koonin, 2006*). All 3 VCs show considerable sharing of PCs within each VC (61.3 %, 50.2% and 67.6%, respectively). VCs 74 and 77, each consisting of 2 genera (*Gammalipothrixvirus*/*Rudivirus* and *Betalipothrixvirus*/*Deltalipothrixvirus*) unify the entire *Ligamenvirales* order (2 families). Though the genera are distinguished mainly by their virion morphology (*Prangishvili & Krupovič, 2012*), it can be argued that some lipothrixviruses share as much similarity within the *Lipothrixviridae* family as to the rudiviruses, exemplified by the 10 genes shared between AFV-1 (a lipothrixvirus) and SIRV1 (a rudivirus) (*Prangishvili & Krupovič, 2012*) and that they likely derive from a common ancestor (*Goulet et al., 2009*). In addition to the number of PCs shared between AFV-1 and the rudivirus in VC74 (Fig. S1), the more "distal" position between AFV-2 (*Deltalipothrixvirus*) and the other VC77 members (*Betalipothrixvirus*) (Fig. S1), the order-level separation is easily seen in the overall network structure (Fig. 2).

VC55 (*Alphafusellovirus*/*Betafusellovirus*) consists of all known *Fuselloviridae* members. Like VCs 74 and 77, their genera are separated mainly through virion morphology, with *Alphafusellovirus* lemon-shaped and *Betafusellovirus* pleomorphic, and also through their attachment structures (*Redder et al., 2009*). The large number of "core" genes (13) shared among all family members argues for frequent recombination events, with even distant fuselloviruses potentially capable of recombination during repeated integration events into the same host. Furthermore, some fuselloviruses exhibit regions >70% pairwise identity on the nucleotide level, including ASV-1 (*Betafusellovirus*) and SSV-K1 (*Alphafusellovirus*) (*Redder et al., 2009*). Despite shared non-core regions between the *fuselloviridae*, the high similarity between the two genera is also revealed in the network through unification into a single VC. The most recently identified member of the *Fuselloviridae*, *Sulfolobales* Mexican fusellovirus 1 (SMF1) has no official ICTV classification between family, though clustering within the VC shows clear association to the *Betafusellovirus*.

## vConTACT, an iVirus tool for network-based viral taxonomy

Given the strong and robust performance of these network classification methods (*Lima-Mendez et al., 2008*) to largely capture known viral taxonomy from genomes alone, we sought to democratize the analytical capability. To this end, we developed a tool named "vConTACT" (overview of its logic in Fig. 1) and integrated it into iVirus, a virus ecology-focused set of tools also known as "apps" and databases (*Bolduc et al., 2016*). Such implementation at iVirus enables any user to run the application simply by providing viral sequences (including novel and/or reference sequences) alongside a CSV-formatted file containing gene and sequence information with all compute, storage and data repository happening via the CyVerse cyberinfrastructure (formerly the iPlant Collaborative (*Goff et al., 2011*). Guides to using vConTACT can be found at dx.doi.org/10.17504/protocols.io.gwdbxa6 (preparing data) and dx.doi.org/10.17504/protocols.io.gwcbxaw (running vConTACT). A pipeline detailing its use alongside other vConTACT-enabled apps is shown in Fig. S5.

## Limitations and future developments of vConTACT

Since vConTACT uses a genome similarity network, it displays the extent of shared genes between genomes as edges, but not what the shared genes are *Corel et al. (2016)*. This lack of information on the identity of shared genes (i.e., host-related genes and ancestral viral genes) in the graph makes the biological interpretation of network connections difficult, and can lead to a misunderstanding of genome evolution (i.e., *T5virus*) when using topology to detect the chimeric viruses. Additionally, the limiting resolution of MCL in poorly-sampled regions of and/or highly- overlapped viral genomes cannot uncover their hidden substructure (i.e., *Cp8virus* and mycophages, respectively). These particular types of limitations had not been reported previously, likely because of the smaller dataset available at the time.

However, we have shown that the combined use of multiple clustering approaches (e.g., MCL and hierarchical clustering) is better able to detect multiscale modularity of the heterogeneous VCs. It is thus possible that more sensitive algorithm(s) can separate the

sub-sampled and/or highly-overlapped genomes from VCs to which they are spuriously assigned and estimation of the statistical significance of VCs can not only distinguish them from other VCs (*Nepusz, Yu & Paccanaro, 2012*), but provide a confidence score for their assignment. Additionally, while a bipartite network is arguably more appropriate to detect mosaic genomes (*Corel et al., 2016*), estimation of in-/out-VC (or genus) cohesiveness may help to characterize the genomes with high overlaps. Thus, although the choices of module detection algorithm and its evaluation are still truly arbitrary (*Fortunato, 2010*; *Schaeffer, 2007*), the application of other approaches should be considered in future work.

## CONCLUSIONS

Network-based approaches have been widely used to explore mathematical, statistical, biological, and structural properties of a set of entities (nodes) and the connections between them (edges) in a variety of biological and social systems (*Dagan, 2011*; *Barberán et al., 2012*). Such approaches are invaluable for developing a quantitative framework to evaluate if and where taxonomically meaningful classifications can be made in viral sequence space (*Simmonds et al., 2017*). We sought here to quantitatively evaluate when and where an existing gene-sharing-based network classification method (*Lima-Mendez et al., 2008*) would perform poorly, and found that only 1 in 4 publicly-available, dsDNA viral genomes were problematic. Follow-up analyses suggested these genomes were problematic due to (i) under-sampled viral sequence space, (ii) incomplete taxonomic assignments of the ICTV genera, and (iii) exceptionally high frequencies of gene sharing between viruses. The ~23% of problematic VCs suffer approximately equally from these issues with 6.5%, 7.5% and 8.4% of the total VCs containing the ICTV genera attributable to each issue, respectively. Fortunately, only the latter group will remain problematic for the approaches presented here as increased sampling of viral sequence space and improvements in network analytics will bring resolution to the former two categories. Thus, three-quarters of publicly-available viral genomes are readily classified via a gene sharing network-based viral taxonomy, and another 14.0% will quickly become so with the remaining ~8% identifiably problematic by network properties and features.

To this end, we present vConTACT as a publicly-available tool for researchers to effectively enable large-scale, automated virus classification. Given thousands of new virus sequences now routinely discovered in each metagenomics study (e.g., *Calusinska et al., 2016*; *Roux et al., 2016*; *Paez-Espino et al., 2016*), and the readiness of the viral community to use genomes as a basis for viral taxonomy (*Simmonds et al., 2017*), these advances take a critical first step towards that goal. Ultimately, only an automatable viral classifier will be able to rapidly and accurately integrate these novel viruses into the meaningful taxonomy so critical for building viruses into predictive ecosystem models across biomes ranging from the oceans and soils to bioreactors and humans.

## ACKNOWLEDGEMENTS

We thank Kate Hargreaves, Consuelo Gazitua, Gareth Trubl, and Dean Vik for testing out beta versions of the vConTACT app, Ann Gregory for review of the manuscript, Bonnie

Hurwitz and Ken Youens-Clark and CyVerse for help implementing the app, and the Sullivan Lab for critical review through the years and comments on the manuscript.

### Funding

This research was supported by awards to MBS from the US Department of Energy, Office of Science, Office of Biological and Environmental Research under the Genomic Science program (DE-SC0010580), the National Science Foundation (OCE-1536989), and the Gordon and Betty Moore Foundation (#3790, 3305) and computational resources provided by the Ohio Supercomputer Center (1987, http://osc.edu/ark:/19495/f5s1ph73). The funders had no role in study design, data collection and analysis, decision to publish, or preparation of the manuscript.

### Grant Disclosures

The following grant information was disclosed by the authors:
Genomic Science program: DE-SC0010580.
National Science Foundation: OCE-1536989.
Gordon and Betty Moore Foundation: #3790, 3305.

### Competing Interests

The authors declare there are no competing interests.

### Author Contributions

- Benjamin Bolduc and Ho Bin Jang conceived and designed the experiments, performed the experiments, analyzed the data, contributed reagents/materials/analysis tools, wrote the paper, prepared figures and/or tables, reviewed drafts of the paper.
- Guilhem Doulcier conceived and designed the experiments, performed the experiments, analyzed the data, contributed reagents/materials/analysis tools.
- Zhi-Qiang You contributed reagents/materials/analysis tools.
- Simon Roux conceived and designed the experiments, analyzed the data, contributed reagents/materials/analysis tools.
- Matthew B. Sullivan wrote the paper, reviewed drafts of the paper.

### Data Availability

Tool source code: https://bitbucket.org/MAVERICLab/vcontact.

### Supplemental Information

Supplemental information for this article can be found online at http://dx.doi.org/10.7717/peerj.3243#supplemental-information.

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
