# Peer review of "vConTACT: an iVirus tool to classify double-stranded DNA viruses that infect Archaea and Bacteria"

_PeerJ, doi:10.7717/peerj.3243_

## Round 0.1 · original submission · Major Revisions

Please carefully address the comments of all reviewers. Specifically, please comment on the challenge that each time new viral data is added to the network the PCs require rebuilding from scratch and consider (or at minimum discuss) the use of existing sets of viral orthologous genes (ie. pVOGs). Please consider (or at least discuss) the use of other module detection algorithms and if using MCL justify the selection of the choice of a particular inflation parameter and evaluate the robustness of the results with respect to different parameter values, as requested by reviewer #1.

Please clarify the gene flow discussion (reviewer #3) and provide details on how to use the vConTACT resource and different versions thereof (reviewer #4).

Lastly, please include additional references as requested by some of the reviewers and ensure consistency and clarity in referring to Figures and providing legend information easily accessible and legible, for both main figures and supplemental material (ie. comments reviewer #2).

Reviewer 1 ·

Basic reporting

1. The article is well written and the language is clear. The introduction provides sufficient background on the recent challenges and advances in the field of virus taxonomy. In that respect, some additional references might be included to illustrate the application of networks to the study of viral phylogeny and taxonomy. For example, Jang et al (J. Virol. 2013, 87(23):12866-78) used a phylogenomic network to assign a newly described phage to the phiKZ group, and Iranzo et al (J. Virol. 2016, 90(24):11043-55) studied the archaeal virosphere and related plasmids using bipartite networks.

2. There seem to be some inconsistencies with the calls to figures in the text. For example, Fig 3B is called from l.317, l.352 and l.356 when discussing VC2, 3, and 14. However, none of those VC appear in Fig 3B. In a similar manner, Fig 4A does not contain VC59, as it would be expected based on its call from l.382.

3. The archaeal subset of Figure 2 would gain from further clarification on the symbols used. For example, in its current form it is unclear whether all circular nodes (e.g. those between VC51 and VC74, or those at the bottom) actually belong to Lipothrixviridae. It would be useful to know what that relatively large connected component to the right of VC74-77 contains. Moreover, I wonder about the location of the archaeal virus members of the Caudovirales, as well as Bicaudaviridae, Turriviridae, Sphaerolipoviridae...

4. Other minor observations:
- Reference (Edwards & Rohwer, 2005) cited in l.89 is missing.
- It would be interesting to see the edge weights represented in Fig 2 and 3.

Experimental design

1. The research question underlying this work is well defined, extremely relevant and meaningful. As explained in the introduction, the use of networks for viral taxonomy is arising as a promising direction that may overcome the limitations of classical approaches. Because of the relevance that this work may have in paving the way for a new approach to viral taxonomy, it is essential that the methodology follows the highest standards. In that regard, the decision to treat the bipartite virus-PC network as a projected (unimodal) genome similarity network seems a bit surprising, if not hard to justify. In the 9 years passed since the publication of Lima-Mendez (2008) new developments in network science have made it possible to analyze bipartite networks directly as they are, without the need of being projected. Specifically, the two network properties exploited in this work (modularity and betweenness centrality) can be now easily studied in bipartite networks. Projection of bipartite networks (i) is arbitrary (i.e. there are several ways to do it and it is not obvious how different procedures affect results); (ii) results in a loss of information (e.g. in the projected network it is impossible to know if two genomes are similar because they share a common set of core genes, a sparse set of horizontally transferred genes, or just genes acquired from the host); and (iii) may introduce structural biases, such as spurious scale free topology (see Montañez et al, Bioessays 2010, 32(3):246-56). As a result, there has been a well justified shift towards studying these networks from a bipartite perspective, not only in the context of phylogenomics (Corel et al, Trends Microbiol. 2016, 24(3):224-37) but also in broader fields such as ecology (see, for example, Flores, Valverde & Weitz, ISME J. 2013, 7(3):520-32) and social sciences (for example Borge-Holthoefer et al, Plos One 2011, 6(8):e23883). That said, the use of a unimodal projection might be, perhaps, justified if the analysis of the network involved algorithms which do not have a bipartite counterpart yet, such as some self-consistent techniques to detect multiscale modularity (see below).

2. The choice of a particular module detection algorithm is naturally subject to some degree of arbitrariness. However, there are some theoretical and practical arguments that make MCL a particularly controversial option for defining viral clusters. From a theoretical perspective, despite MCL finds sets of nodes that look like “reasonable” modules, such sets do not comply with any rigorous definition of what a module must be. As a result, there is no consistent way to assess if the modules are “good” or if one possible partition is better than another. This is a serious drawback, especially if one wants to evaluate the statistical validity of a module (i.e. how unlikely it is that a similar module appears by chance in a random network with the same degree distribution). From a practical perspective, the value of the inflation parameter is an arbitrary choice, with different values typically leading to modules of different size. In the context of viral taxonomy, it is possible that different taxonomical levels are recovered by using different values of the inflation parameter (unfortunately, because there is no rigorous theoretical basis for the algorithm, there is no guarantee that modules obtained with different parameter values are consistent subsets of each other). In any case, if MCL is used as the module detection algorithm, the choice of a particular inflation parameter should be well justified, and the robustness of the results with respect to different parameter values should be evaluated. Other module detection algorithms, such as those based on the Newman’s modularity index (Barber’s modularity for bipartite networks) or on information theory are free of these problems and allow interesting extensions, such as the possibility to find hierarchies of modules at different resolutions in a consistent way. Obviously, the latter possibility would be especially interesting for taxonomic applications. It is also possible (e.g. OSLOM algorithm) to evaluate the statistical significance of nodes within a module in order to discard spurious modules or nodes assigned to the wrong module. For an extensive review on the available methods (up to the date of their publication), see Fortunato, Phys. Rep. 2010, 486(3-5):75-174 and the more recent but less comprehensive Fortunato & Hric, Phys. Rep. 2016, 659:1-44.

3. A desirable property of the PCs is that they should be easily extendable to accommodate proteins from newly discovered viruses. However, based on the definition of PCs as the straightforward outcome of the module detection algorithm applied to the sequence similarity network, it seems that every time new viruses are added to the network the PCs have to be rebuilt from scratch. This drawback could be avoided by using existing sets of viral orthologous genes, such as the pVOGs (Grazziotin, Koonin & Kristensen, Nucleic Acids Res. 2017, 45:D491-D498), so that new sequences can be added in a consistent way based on profile comparisons with the existing orthologous sets.

4. Other minor points:
- Please define or explain the meaning of the terms in equations (1) and (3).
- The sentence in l.194 is hard to understand.
- It says in l.339 that the number of genomes of a given group was doubled computationally. The meaning of this sentence is not clear to me. Does it mean that replicas of those nodes were added to the networks? What was the weight of the links between the original nodes an its replicas?

Validity of the findings

1. Despite the methodological caveats described above, the main conclusion of the work, namely that viruses can be studied as a network whose modules represent taxonomically meaningful units, seems reasonable. However, some specific results may be partly derived from a lack of resolution of the algorithms used. For example, the article describes some instances of “lumping” that can be resolved by hierarchical clustering of the affected VC. It is thus possible that more sensitive module detection algorithms can split those groups in a more precise way. Similarly, it is possible that some of the viruses that "do not seem to belong" to the VC to which they are assigned were removed if the statistical significance of nodes within modules were assessed.

2. Concerning the discussion on the betweenness centrality of Mycobacterium phages, it should be noted that the betweenness centrality in a unimodal genome similarity network must be interpreted carefully. For example, based on the high betweenness of a single representative of T5-like phages, Lima-Mendez et al (2008) concluded that T5 is a mosaic virus that acts as a bridge between T4-like viruses and lambda-like viruses. The subsequent analysis of the bipartite virus-PC network carried out by Iranzo, Krupovic & Koonin (2016) revealed that most of the core genes shared by T5-like viruses and lambda-like viruses are widespread viral hallmark genes, which are also present in T4-like viruses. Therefore, the apparent bridge location of T5-like phages in the unimodal genome similarity network was likely an artifact derived from the loss of information in the projected network, together with the incomplete sampling of the T5-like viruses. As pointed out by Corel et al (Trends Microbiol 2016, 24(3):224-37), the most suitable approach to the detection of mosaic genomes would be the study of the bipartite network.

3. More specific comments:
- The statement “It is remarkable that all known archaeal viruses not only fall within the network, but that most of their VCs follow a genus-level affiliation” (l.454) seems inconsistent with the statement “archaeal virus were incorrectly classified at the genus level” (l.292).
- The subsection title in l.293 seems inappropriate, as long as it is later stated that some VCs include viruses that should be assigned to a different VC (e.g. Eucampyvirinae from VC1 to VC87). Perhaps “Gene content analyses suggest ICTV classifications should be revised” would be a better title.
- It would be interesting to see a plot similar to Fig 2C, but showing the statistics about the number of VCs in which ICTV groups split.

Reviewer 2 ·

Basic reporting

No comment.

Experimental design

No comment.

Validity of the findings

No comment.

Additional comments

This is an interesting paper that uses proteomic similarities to explore viral (primarily phages and viruses of Archae) taxonomy.

In general, the approach of using proteomic analysis in which the predicted encoded proteins are assorted into related groups (PCs) and the distribution among genomes is compared, is not new per se, and mirrors approaches used by others. However, the analysis is more sophisticated and more robust, and has been incorporated into an online tool that can be used by the community if they wish to get a report on the taxonomy of any newly sequenced genome. In general, this makes for an interesting and informative analysis, especially when compared with alternative taxonomies such as that proposed by the ICTV.

The paper goes into considerable details on discrepancies between the VCs reported here and the ICTV classifications. On line 267 it states that 76.4% of the 211 VCs contain a single ICTV-accepted genus, concluding good concordance between the taxonomies. However, this also indicates a substantial level of discordance, and although the basis for this is discussed briefly, some further explanation would be helpful. For example, there is a discussion of the VCs with 2 or more ICTV genera, but not much explanation as to the basis of this discordance.

Although the paper is overall well-written, some parts are quite impenetrable. This is exacerbated because of the apparent lack of legends for the Supplementary figures. Perhaps they are in there somewhere, but I couldn’t find them! For example, Fig. S2 shows the mycophages with high BC values, but the figure is difficult to interpret. Why are some ‘high’ BC genomes labeled and not others. What is the x-axis? What VC(s) are the labeled phages in? This was not easy to figure out from the Supplementary Tables. What is the explanation for these particular phages having these characteristics? Could it be related to mobile genetic elements in part that are more broadly distributed than other genes?

For the enterophages and mycophages there have been prior attempts to group phages according to relatedness using other measures including gross nucleotide similarity (see Grose & Casjens, 2013; Pope et al., 2015). It would be of interest to determine the concordance of the VCs with the clusters identified using those methods. In the prior studies, there is also some consideration of other hierarchical levels, including super-clusters and sub-clusters. How do the VCs and the other parameters such as BC vlaues compare to these? These comparisons would strengthen the paper and the utility of the analyses.

The manuscript discusses some of the profiles that are seen with phages having different degrees of mosaicism. The underlying reasons are perhaps obscure, but it is worth noting that the relatively large group of sequenced mycophages share the property of being able to infect the same host strain, a characteristic that is rare among other phage groups. Thus the relationships reported among the mycophages could be a rare oddity, or they could reflect a feature that is common among phages but which is not apparent because of the variety of host strains used for propagation. This does not appear to be the case for Synechococcus phages, but how these observations relate more broadly elsewhere in the phage community remain obscure. This is perhaps worthy of some discussion, while also noting the need for expanding the very limited current collection of sequenced phage genomes.

The paper is clearly data-rich, but the small labeling on many of the figures and supplemental information make them difficult to read.

In Figure S1, Min1 is incorrectly labeled as a mycobacterium phage, whereas in Table S1 it is correctly identified as a Microbacterium phage. Perhaps this was a result of a manual adjustment, and if so, some additional proofreading might be helpful.

Reviewer 3 ·

Basic reporting

no comment - all good.

Experimental design

I felt the others experimental design was well thought out and examined during various stages of analysis.

Validity of the findings

no comment

Additional comments

The authors describe a new method for the binning and taxonomic assignment of dsDNA viral sequences called vconTACT. It is a clever approach that will aid the field of viral ecology. They showed in a robust way, bioinformatically, the need for increased sampling of phage groups/clusters where the resolution or similarity to the ICTV classification was not attained. This has been seen in phylogenetics and shown again here using their method.

Major:
Lines 50-74: I’m confused by the approaches listed. While they are valid and great examples they are not inclusive as it might be assumed by the reader. Also, it was unclear if the focus was on the computational approaches as stated at Line 50. Mizuno 2013 – a great study, is a string of smartly organized previously described computational tools. Also there are other studies to list in some of these categories. And, the authors, I believe, are targeting their introduction on marine viromes and this should be stated as there are many approaches within infection disease fields and other human-associated microbe/viral areas. I am also curious why the authors chose to focus so much on the marine environment in the introduction (although I realize this is the corresponding authors main environment), when vconTACT was not evaluated with environmental sequences in this study. Perhaps slight rewordings and compaction of this section of the intro could help clarify.

Line 77-80: The authors discuss gene flow – the transfer of genes. Are they referring to genes shuttled between viruses or between viruses and their hosts? Or is the flow of genes from one population that mixes with another population. Since these are non-sexual organisms/entities the exchange of alleles through classical interbreeding would not be a process. Please clarify. When discussing gene flow … I understand why temperate phage may be more susceptible to gene flow possibly during ‘sloppy’ extraction from the host genome during viral replication, however, during lytic infection could viruses also pick up genes from degraded host genomes etc? I’m not as familiar with this literature as the authors likely are. Could they comment and also provide references for the statements that lytic would be less likely to evolve as a result of gene flow. Also, the statement that dsDNA viruses might not be a susceptible to evolution through gene flow compared to the fast evolving RNA and ssDNA viruses should be referenced and clarified. Fast evolution of RNA and ssDNA is usually due to higher mutation rates (as compared to dsDNA phage and other cellular organisms), not necessarily gene flow.

Gregory 2016 was cited at Line 278, which does discuss physical differences and also states “Mechanistically, while gene flow in bacteria and archaea can occur from multiple sources, gene flow in viruses is restricted to times when two viruses co-infect the same host. Such “co-infection” need not be simultaneous, but it does require spatial proximity and shared host range.” So throughout this manuscript gene flow refers to co-infected phage. Again more clarification is needed.

The section from Lines 414-424: appears to not fit within the Autographivirinae section, perhaps I missed something. Also, I don’t see VC22 in Figure 4B as referenced. VC 2 is there but has only Staph phage, I believe – the resolution was a little low.

Could the authors draw any conclusions about how to discern ‘lumping’/gene flow/mosaic genomes using bioinformatics (where no ICTV or other references are present, i.e., from purely environmental bins) based on this study? And, perhaps this should be discussed in the conclusions.

Minor comments:

Line 57: “These studies have added 15,222 and 125,842 new virus sequences …” I realize based on the previous sentences that these are genomes and large genome fragments, but using sequences could be confused with reads and genes which is obviously much more. Perhaps consider merging the two sentences or providing more specific characterization of the sequences.

Line 102: following iii) consider revising for clarity, “which for the oceans at least represents <1% of the viral genomes recovered”. I believe the authors are trying to state that our current studies have only uncovered <1% of the viral genomes predicted to present in the worlds
oceans.

Line 108: MCL – write this out, Markov cluster.

Line 110: should be represent instead of represents.

Line 274: perhaps consider “blurring” to ameliorates.

Line 396: Should VC_8 be VC8 like the previous used nomenclature for the VCs. Same for VC_1 at Line 397. Also here, should T-evens/Pseudo/Schizo T-evens be Pseudo/Schizo T-evens?

Line 444: The names should be italicize - Alphafusellovirus and Betafusellovirus.

Line 445: I’m confused by the following sentence, “The large number of “core” genes (13) shared among all family members argues for frequent recombination events, with even distant fuselloviruses potentially capable of recombination during integration.” Do the authors mean recombination during integration as in multiple viruses integrate the host genome during the same infection event? Please clarify.

Reviewer 4 ·

Basic reporting

I enjoyed reading the present article "vConTACT" in which the authors use a whole-genome approach (via protein content) to taxonomically classify known microbial viruses (from the reference database) and compare their approach with the current system provided from the ICTV. Then, this tool is integrated within the Cyberse cyberinfrastructure for the public to use it.

This is essentially an extended version of a previous work from the authors, where now, they include a higher number of reference viral proteins derived from a higher number of viruses and examine the inconsistencies between their method and the current taxonomy (there is a ~24% of non-concordance at the genus level).

I am curious to know how much "vConTACT" has changed since the iVirus publication (July 2016).

The article is well structured, detailed in the literature, figures are informative and with nice aesthetics, and the results seems sound.

I was, however, very confused with the use of the tool itself and I think that the article needs to add more clarification and details on how to use this resource. There are several versions of vConTACT in the "Cyverse Discovery Environment." Then, assuming that version 0.1.60 is the correct, there is no explanation on how to proceed to analyze users viral sequences. There are 2 other "Apps" for vConTACT: PCs (2 versions of it) and "genes2contig." The information button provides no information. This should be well addressed.

The Introduction section sets very well the current problem and the need of a new/complementary viral taxonomy in this big data era. However, the authors refers here and there to viruses in a general term and should be a bit more precise when they refer at least to DNA viruses or RNA viruses, which are very different. For instance, the 3 approaches they describe (lines 51-62) are discovering exclusively DNA viruses, and the 2 tools described (lines 92-98) are tested in RNA viruses. These RNA/DNA terms should be included to set up a better context of what is being done.

In Lines 25 and 120, the authors mentioned that the current tools is "optimized". Please add how it is optimized.

Line 57, the reference Paez-Espino, 2016 should be updated (or complemented) with Paez-Espino NAR 2017 where they add ~250k viral sequences.

Line 119, can the authors say up to what taxonomic level they are confident to assign?

Line 145, In this method section details of the BLASTp need to be included.

Line 153, when is considered significant?

Line 250, how many singletons were found?

Line 267, if 76.4% of the data agreed at the genus level with ICTV the authors should say that "Thus, roughly 3 out of 4 of the VCs corresponds to ICTV genera" in line 270.

Line 288, to be consistent with the whole sentence, add here the family of the VC2 (I think it is Siphoviridae).

Line 290, same thing here than the previous comment (add family ... Myoviridae?).

Table1, VC: what is a significant number of genes? can a percent be added?

Figure3, Caption should include the color code and not refers to Figure2.

Figure4B, Viral species are illegible.

Experimental design

The work, although not original, represent a considerable increase with respect to the previous work in 2008.

The work seems very well defined and the authors have done a very good job investigating deeply the inconsistencies.

A couple of questions here:

Is the tool only based on PCs shared or does it also consider some gene synteny?

The authors mentioned that some microbial "cargo" genes may lead to a wrong taxonomic assignment, have the authors considered removing microbial gene hits?

Validity of the findings

Have the authors considered testing some of the available massive viral databases that they enumerate in the Introduction section (i.e. Tara Oceans, Prophages, Earth Virome ...?

If so, where they able to assess a comparable taxonomy with the one reported (if any)?

---

## Round 0.2 · accepted · Accept

We look forward to publishing this nice contribution.